# CONTROLLABLE TEXT-TO-IMAGE GENERATION VIA AUTOMATIC SKETCHES

## ABSTRACT

Current text-to-image generation models often struggle to follow textual instructions, especially the ones requiring spatial reasoning. On the other hand, Large Language Models (LLMs), such as GPT-4, have shown remarkable precision in generating code snippets for sketching out text inputs graphically, e.g., via TikZ. In this work, we introduce Control-GPT to guide the diffusion-based text-to-image pipelines with programmatic sketches generated by GPT-4, enhancing their abilities for instruction following. Control-GPT works by querying GPT-4 to write TikZ code, and the generated sketches are used as references alongside the text instructions for diffusion models (e.g., ControlNet) to generate photo-realistic images. One major challenge to training our pipeline is the lack of a dataset containing aligned text, images, and sketches. We address the issue by converting instance masks in existing datasets into polygons to mimic the sketches used at test time. As a result, Control-GPT greatly boosts the controllability of image generation. It establishes a new state-of-art on the spatial arrangement and object positioning generation and enhances users' control of object positions, sizes, etc., nearly doubling the accuracy of prior models. Our work, as a first attempt, shows the potential for employing LLMs to enhance the performance in computer vision tasks.

## 1 INTRODUCTION

Researchers have made remarkable progress in text-to-image generation in recent years, with significant advancements in Generative Adversarial Networks (GANs) Brock et al. (2019); Goodfellow et al. (2014); Karras et al. (2019), autoregressive models Ramesh et al. (2021b); Yu et al. (2022), and Diffusion Models Ramesh et al. (2022); Saharia et al. (2022); Rombach et al. (2022). These models have shown impressive capabilities in synthesizing photorealistic images. More recently, large-scale pretrained text-to-image models Ramesh et al. (2021a); Rombach et al. (2022); Saharia et al. (2022) have garnered considerable interest, leading to a multitude of downstream applications, such as image editing Brooks et al. (2023); Hertz et al. (2022) and image inpainting Rombach et al. (2022).

However, precise control during image generation from textual inputs remains a formidable challenge Gokhale et al. (2022). As illustrated in Figure 1, specifying the exact location, size, or shape of objects using natural language is inherently difficult and prevalent models like DALL-E 2 Ramesh et al. (2022) or Stable Diffusion Rombach et al. (2022) often lead to unsatisfactory results. To address this problem, existing approaches often depend on extensive prompt engineering Rombach et al. (2022) or manually created image sketches Zhang and Agrawala (2023); Li et al. (2023). They are both inefficient and difficult to generalize, as they demand substantial manual effort.

On the other hand, users typically have greater control over code generation, as they can write programs to manipulate various aspects of objects, such as their shapes, sizes, locations, and more. This allows for precise adjustments and customization according to specific requirements. In recent times, large language models (LLMs) have demonstrated remarkable capabilities in code generation tasks. Models such as GPT-4 OpenAI (2023) have achieved near-human-level performance in coding contests and have been successful in solving numerous complex coding problems Bubeck et al. (2023); Lu et al. (2021). This progress encourages us to investigate the potential of harnessing LLMs to enhance the controllability of text-to-image generation models.

In this work, we introduce Control-GPT, a simple yet effective framework that harnesses the power of LLMs to generate sketches based on text prompts, illustrated in Figure 2. Control-GPT first employs

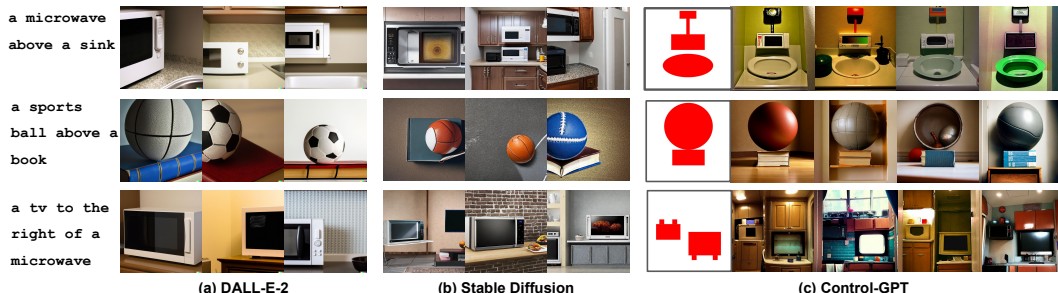

a microwave above a sink

a sports ball above a book

a tv to the right of a microwave

(a) DALL-E-2      (b) Stable Diffusion      (c) Control-GPT

Figure 1: **Controllable text-to-image generation with GPT-4 in the loop**. Among all the models, Control-GPT is both good at generating multiple objects, and the generated image follows exactly the TikZ sketch. Both DALL-E 2 and Stable Diffusion are not only unable to generate all the objects stated in the text consistently, but it is also hard to control their generated image layout.

GPT-4 to generate sketches in the form of TikZ code, inspired by Bubeck et al. (2023). As we can see from Figure 1 (c), the programmatic sketches follow the exact text instructions despite the rareness of the concepts among natural images. Subsequently, these sketches are fed into Control-GPT, a variant of Stable Diffusion that accepts additional inputs, such as reference images, edges, and segmentation maps. These generated sketches act as reference points for diffusion models, enabling them to understand spatial relationships and unusual concepts better, as opposed to relying solely on text prompts. This process eliminates the need for human intervention in prompt engineering or sketch creation and helps improve diffusion models' controllability.

In practice, utilizing off-the-shelf pretrained models to synthesize images from textual prompts and TikZ sketches can often produce unsatisfactory results. For example, GPT-4 generated sketches achieve close to 97% accuracy in following the spatial relations described in Gokhale et al. (2022). However, directly using the generated sketches as segmentation maps and feeding them to the pretrained ControlNet cannot translate the performance improvement. This is due to text-to-image models possibly lacking a full understanding of the content within TikZ sketches, as the pretrained models haven't seen such sketches during training. To address this issue, we convert instance masks from existing instance segmentation datasets, such as COCO Lin et al. (2014) and LVIS Gupta et al. (2019), into polygon representations, which are similar to sketches generated by GPT-4. We construct a dataset containing triplets of images, textual captions, and polygon-represented sketches and finetune ControlNet, which takes additional polygons and grounding tokens with object names and locations Li et al. (2023) on the constructed dataset. This approach can mitigate the gap in understanding the GPT-generated sketches and help the model better follow the text prompt instructions.

For evaluation, we first investigate the capabilities of widely-used LLMs (e.g., LLaMA Touvron et al. (2023) and Alpaca Taori et al. (2023) in addition to the GPT series of models) in the context of sketch generation. We then evaluate our image generation framework, which incorporates GPT-4, on the spatial relation benchmark created by Gokhale et al. (2022). Our proposed model sets a new state-of-the-art with an accuracy rate of 44.2%, almost doubling the performance of the standard stable diffusion models (18.8%). Moreover, human evaluation reveals that our model can handle out-of-distribution prompts and generate intricate scenes comprising multiple objects. As an initial effort, our work demonstrates the potential of integrating the coding abilities of LLMs within visual domains to enhance model controllability.

## 2    RELATED WORK

**Text-to-image models.** Large-scale text-to-image models, which are categorized into autoregressive models Yu et al. (2022), and diffusion based models Ramesh et al. (2021a); Rombach et al. (2022); Saharia et al. (2022) have shown impressive results. The diffusion-based model has recently attracted increasing attention for text-to-image generation. Text-to-image diffusion models often work by encoding text prompts into latent vectors and then using a diffusion model to diffuse pixels. Notable large-scale models include Stable Diffusion Rombach et al. (2022), Imagen Saharia et al. (2022), DALL-E Rombach et al. (2022), etc. We take diffusion-based text-to-image generation models as a basis and explore how to increase their controllability for precise instruction following.

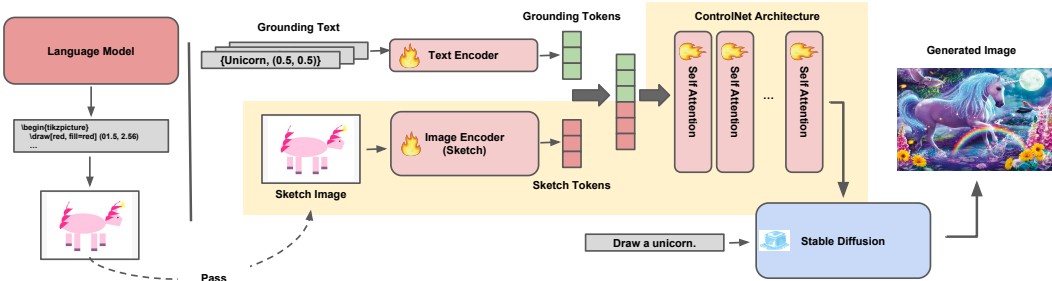

Figure 2: **Control-GPT Architecture**. Our model is built on top of ControlNet to take additional grounding text. The model takes in both reference images and grounding object text, fusing them using attention layers before feeding to Stable Diffusion.

**Controllable generation.** Current state-of-the-art image diffusion models predominantly focus on text-to-image methods, making text-guided approaches the most direct method for enhancing control over diffusion models. Achieving precise control via text is inherently challenging. ControlNet Zhang and Agrawala (2023) is a neural network architecture that controls pretrained large diffusion models to support additional input conditions like Canny edges, Hough line, human poses, segmentation maps, etc. This model design enjoys greater flexibility of the condition signals and allows for more precise control of the generated images. However, such condition signals often need additional human effort to obtain and may not be available for open-world concepts. Our work addresses this issue by integrating diffusion models with LLMs to generate sketches at test time automatically.

In a similar line, GLIGEN Li et al. (2023), an approach for open-set grounded image generation, adds an additional grounding token to encode the bounding box coordinates of the objects. Our work complements GLIGEN and incorporates its grounding token design. Our work is also in line with image generation from layouts Zhao et al. (2019); Sun and Wu (2019; 2021); Jahn et al. (2021); Li et al. (2021). These techniques typically operate in a closed-world environment characterized by a fixed vocabulary and access to predefined layouts.

**Programmatic sketch generation.** Large language models (LLMs) have exhibited remarkable performance on text-related generation tasks OpenAI (2023), and GPT-4 stands out as the most advanced LLM to date. Recently researchers have shown that GPT-4 exhibits "sparks" of human-level intelligence Bubeck et al. (2023) in a range of application domains. In particular, GPT-4 exhibits near-human level coding capability as well as an unexpected surprise —its "image generation" via writing code, where examples include GPT-4 successfully writing a snippet of TikZ commands in LaTeX that draws a "unicorn" graphic. However, the resulting images are usually far from photo-realistic. Our work further quantifies GPT-4's programming ability on sketch generation and integrates it into diffusion-based image generation models for better controllability. Our work, as a first attempt, shows the potential of integrating LLMs with other domains with a code interface.

## 3 CONTROLLABLE TEXT-TO-IMAGE GENERATION WITH GPT-4

### 3.1 PRELIMINARY: CONTROLNET

ControlNet Zhang and Agrawala (2023) is a variant of diffusion models that takes additional input conditions. Built upon the Stable Diffusion model, ControlNet adds a conditional pathway, a trainable copy of the 12 encoding blocks, and one middle block of Stable Diffusion while locking the parameters of the trained Stable Diffusion model. The neural network blocks are connected by a special convolution layer, "zero convolution", which is a $1 \times 1$ convolution layer with both weight and bias initialized with zeros.

During training, ControlNet is finetuned on image-text pairs with conditions represented as Canny edges, Hough lines, human poses, user sketches, semantic segmentation maps, etc. Experiments show that ControlNet can generate images following the conditions. In this work, we use ControlNet as the base image generation model and extend it with pathways for programmatic sketches and grounding tokens, detailed in the following section.

## 3.2 Our Framework

We now introduce our framework that integrates LLM-generated sketches into ControlNet for more precise controllability. As illustrated in Figure 2, we first prompt GPT-4 to generate sketches in TikZ code following the text descriptions and additionally output the object positions. We compile the TikZ code in LaTeXand convert the sketches into image formats. We then feed the programmatic sketches, text descriptions, and grounding tokens of object positions to a tuned ControlNet model to generate images following the conditions.

Training a ControlNet with GPT-4 generated sketches is necessary as pretrained ControlNets do not understand the generated sketches and cannot translate them into realistic images. The challenge is that there are no aligned text, images, and sketches data available for training. In the following sections, we will describe the components in our framework and our training process to mitigate the gap in understanding the programmatic sketches.

### 3.2.1 Training Data Construction

Feeding the Language Model-generated Tikz output directly into ControlNet enhances the controllable generation process. However, ControlNet often struggles to accurately interpret the sketches provided. Common issues include the model failing to adhere to the layout of the sketch or the prompts given. Therefore, fine-tuning ControlNet on sketches is necessary for the model to accept the guidance from programmatic sketches at test time.

When fine-tuning text-to-image models, researchers often use massive datasets of text and image pairs from the Internet (e.g., Flicker Plummer et al. (2015), LIAON-5B Schuhmann et al. (2022)). In our case, although GPT-4 can synthesize unlimited sketches, it's hard to find ground-truth images that align with the sketches. Therefore, we explore methods that utilize annotations from existing image datasets and try to mimic the sketches to mitigate the gap between the training and testing data.

Two zebras seem to be embracing in the wild

A stove with a lighted hood in the kitchen

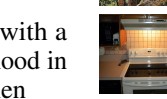 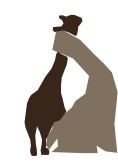 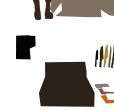

(a) Captions  (b) Images  (c) Polygon Sketches

Figure 3: **Training data construction.** We convert the instance masks in LVIS data into polygons and use the corresponding images and captions from COCO to construct the training data to fine-tune ControlNet.

Specifically, we find that representing instance masks in polygons and turning them into sketch-like images can reduce the gap. As shown in Figure 3, we construct a dataset of caption, image and polygon sketch triplets from the COCO Lin et al. (2014) and LVIS Gupta et al. (2019) dataset and use it to fine-tune ControlNet. LVIS and COCO share the same set of images and LVIS provides a larger object vocabulary of over 1200 classes than COCO. We take roughly 120K images and captions from COCO and convert the object masks from the corresponding LVIS annotations. This enables the model to take in the polygons representing objects in an image and generate an output that adheres to the desired style. We will release our constructed dataset to the community.

### 3.2.2 Additional Object Grounding

Complementary to sketches and text instructions as a condition, we find that adding additional object grounding tokens to ControlNet is useful to further eliminate ambiguity. The grounding token is defined to associate the sketches with object names and positions, serving as an explicit semantic label for the sketches. This technique is particularly helpful when the LLM-generated sketches might be too abstract and not accurately represent the intended objects (Figure 4). Without the grounding token, we can see generation errors like the model successfully follows the layout of the provided sketch but generates incorrect objects. Detailed ablations can be found in Appendix.

To incorporate the grounding tokens in ControlNet, we design grounding tokens in the form of `{object_name, object_position}`:

$$\text{Grounding Tokens}: \mathbf{l}_{\text{gd}} = \{l_{\text{o}}, l_{\text{p}}\} \tag{1}$$

$$\text{Image Tokens}: \mathbf{l}_{\text{img}} = \{l_{\text{patch}_1}, l_{\text{patch}_2}, ..., l_{\text{patch}_k}\} \tag{2}$$

$$\text{All Control Tokens}: \mathbf{l} = \text{Concat}(\mathbf{l}_{\text{gd}}, \mathbf{l}_{\text{img}}) \tag{3}$$

Here, $l_{\text{o}}$ represents the object name grounding token, $l_{\text{p}}$ denotes the position embedding token, and $l_{\text{patch}_i}$ denotes the $i_{\text{th}}$ image patch token in the visual transformer outputs, as illustrated in Figure 2. We permit a maximum of 30 tokens per image and feed these tokens, along with the image tokens, into the transformer blocks to generate control signals.

**Fine-tuning process.** During training, we fine-tune ControlNet with our constructed dataset based on COCO. We freeze the stable diffusion blocks and the text branch in ControlNet while fine-tuning the control branch of the network with all the control tokens $\mathbf{l}$, a concatenation of image tokens, and grounding tokens. This design is flexible and effectively separates the control branch from the general stable diffusion, allowing for more flexible updates on either side.

### 3.2.3 QUERYING GPT-4 FOR PROGRAMMATIC SKETCHES AT INFERENCE

Once the ControlNet is trained, we can now integrate the GPT-4 pipeline into the text-to-image generation process. Beyond the objects and scenes that are trained, users can query GPT-4 with novel prompts in a zero-shot manner and request Control-GPT to generate photo-realistic images.

To prompt GPT-4, we ask users to follow the prompt example below, which will request GPT-4 to request structured outputs of TikZ code snippets, and the associated object names and positions. We can then use the output from GPT-4 to compile sketch images and obtain the grounding tokens. Note that we specify the image size as 5.12-by-5.12 to make sure LLMs do not get confused by large values. We later scale the output by $100\times$ to match the size of the COCO images used at training. We will present examples of the TikZ code in Appendix and release the code together with the rest of the dataset to the community.

```
1  "input": "Draw a tv above a surfboard using TikZ without adding labels.
       The entire image should be inside a 5.12*5.12 bounding box. First,
       you need to provide a step-by-step drawing guide. Then, you need to
       generate the code following the guide. Finally, summarize the drawing
        with: Summary of the drawing, {'object name': $OBJECT_NAME, '
       position': $(X, Y)} Make sure each object is separated and filled with
        red color."
```

## 4 HOW ACCURATE ARE LLMS AT DRAWING SKETCHES?

The precision of Control-GPT largely relies on the precision and controllability of LLMs at generating the sketches at first hand. In this section, we conduct a human evaluation of the outputs of prevalent LLMs and benchmark the performance of LLMs on sketch generation. We find that the GPT-series models are significantly better than open-sourced models like LLaMa Touvron et al. (2023) and Alpha Taori et al. (2023) on sketch generation and GPT-4 exhibits astonishingly high accuracy ($\sim$97%) at following text instructions.

**Human evaluation.** We randomly sample 100 queries from the Visor Dataset Gokhale et al. (2022), which includes 25K text prompts specifying the spatial relationships of two objects like "a carrot above a boat", "a bird below a bus". These text prompts are challenging partially because many of them are rare compositions of two unrelated objects, and associating the spatial deixis in text with regions in images is not easy. Shown in Table 2 in Section 5, DALL-E 2 and Stable Diffusion can only achieve 37.89% and 18.81% on this benchmark. We will defer detailed comparison later.

| Models | # of errors w.r.t instructions | # of empty image or non-runnable code |
|---|---|---|
| GPT-4 OpenAI (2023) | 3 | 5 |
| GPT-3.5 | 24 | 7 |
| ChatGPT Cha (2022) | 60 | 7 |
| LLaMA Touvron et al. (2023) | - | 100 |
| Alpaca Taori et al. (2023) | - | 100 |

Table 1: Human evaluation on sketch generations.

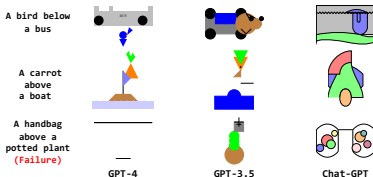

Figure 4: Visualization of the generated sketches

| | Control-GPT | GLIGEN | GLIDE | GLIDE+CDM | DalleM | CV2 | DALL-E 2 | SD | SD+CDM | Δ |
|---|---|---|---|---|---|---|---|---|---|---|
| Uncond (%) | **44.17** | 13.76 | 1.98 | 6.43 | 16.17 | 12.17 | 37.89 | 18.81 | 14.99 | 25.36 |
| Cond (%) | **65.97** | 31.41 | 59.06 | 63.21 | 59.67 | 65.89 | 59.27 | 62.98 | 64.41 | 2.99 |
| OA (%) | 48.33 | 19.78 | 3.36 | 10.17 | 27.10 | 18.47 | **63.93** | 29.86 | 23.27 | 18.57 |
| Visor 1 (%) | 69.80 | 35.64 | 62.86 | 70.52 | 71.22 | 74.37 | 75.67 | 74.89 | | -5.09 |
| Visor 2 (%) | **51.20** | 13.80 | 9.54 | 16.48 | 32.54 | 25.40 | 53.86 | 32.67 | 27.65 | 18.53 |
| Visor 3 (%) | **35.67** | 4.55 | 1.59 | 2.92 | 12.82 | 7.15 | 26.52 | 11.19 | 9.19 | 24.48 |
| Visor 4 (%) | **20.48** | 1.06 | 0.26 | 0.40 | 3.65 | 1.26 | 8.54 | 2.65 | 2.12 | 17.83 |

Table 2: **Results on the Visor spatial dataset**. We compare different image generation models on their abilities to understand spatial locations. Results show that Control-GPT has a significant win on unconditional and conditional accuracy, even compared to the closed-source models. For object generation accuracy, Control-GPT also reaches a SoTA performance (except for DALL-E 2, a model trained in private data).

We examine the performance of LLMs by querying them to generate a TikZ code snippet to describe the given text description. As shown in Table 1, most of the code snippets from GPT-series models can be compiled to valid sketch images, while the outputs from LLaMA and Alpaca are either empty or not runnable. Within the GPT-series models, we find the latest GPT-4 only fails three times out of the 95 queries, which have succeeded in generating a valid sketch, giving a roughly 97% in successfully following the text instruction. ChatGPT, a finetuned version of GPT-3.5 with reinforcement learning with human feedback (RLHF), significantly underperforms the original GPT-3.5. There might be a trade-off between the chatting ability and the code generation during the tuning process.

**Illustration.** In Figure 4, we provide a visualization of sketch examples from the GPT-series models. Although the generated sketches are not photo-realistic, they often capture the semantic meaning and correctly reason the spatial relationship of the objects. The generated sketches often surprisingly get object shapes correct with simple code snippets. In the last row of the figure, we show one of the failure cases of GPT-4, where the model fails to generate the object shape, while GPT-3.5 manages to give a correct sketch. The high precision of GPT-4 at sketch generation encourages us to use it to enhance the controllability of image generation models.

## 5  EXPERIMENTS

In this section, we evaluate Control-GPT on a range of experimental settings to test its controllability regarding to spatial relations, object positions, and sizes based on the Visor dataset Gokhale et al. (2022). We also extend the evaluation to multiple objects and out-of-distribution prompts. Extensive experiments show that Control-GPT can significantly boost the controllability of diffusion models.

**Baselines.** We compare with a wide range of baselines, including open-source and closed-source models. Open-sourced models include (1) *GLIGEN* Li et al. (2023): a fine-tuned Stable Diffusion Jahn et al. (2021) model that takes bounding box and object name as additional control information for grounded text-to-image generation; (2) *ControlNet w/Segmentation* Zhang and Agrawala (2023): a variant of the Stable Diffusion model using instance segmentation maps as control information; (3) *ControlNet w/ Canny Detection*: a variant of ControlNet using canny edge detection images as condition; (4) *Stable Diffusion*: an open-source image generation model conditioned on text trained on LAION-5B dataset (Schuhmann et al., 2022) without additional conditions except for text; (5) *CogView2* Ding et al. (2022): a pretrained image generation model via hierarchical transformers; and (6)*GLIDE* Nichol et al. (2021): a text-conditioned image generation diffusion model by OpenAI. For the closed-sourced model, we compare with (7) *DALL-E 2* Ramesh et al. (2022): OpenAI's most advanced model for image generation. Our model is based on ControlNet, and we use DALL-E 2 for benchmark comparison only.

**Dataset.** As mentioned before, our model is finetuned on COCO (Lin et al., 2014) images and captions with LVIS (Gupta et al., 2019)'s instance annotations. We extract the polygon of each object in the dataset and use the bounding box to generate the grounding tokens. The size of our dataset is approximately 120k images and captions. We finetune the model with four epochs of a learning rate 2e-5. We also freeze the Stable Diffusion backbone and only finetune the control branch in the network. The details for training can be found in Appendix.

### 5.1  FOLLOWING SPATIAL RELATIONS IN TEXT PROMPTS

We adopt the Visor benchmark defined in Gokhale et al. (2022) to examine the models' ability to follow spatial relations in the text instructions. We follow Gokhale et al. (2022) for the evaluation

metrics. Given an object A, an object B and their spatial relationship in text, Object Accuracy (OA) is used to determine whether the model can generate both objects A and B. Unconditional Accuracy (Uncod) is used to examine whether the model can generate object A and B, as well as get their spatial relationship correct. Conditional Accuracy (Cond) is conditioned on object A and B being correctly generated and determines whether the model generate their spatial relationship. Visor $k(k = \{1, 2, 3, 4\})$ means for a given relationship, at least $k$ generation has the correct object and relationship. For each text prompt, we sample four images to make the evaluation more consistent.

We present the evaluation results in Table 2. The Control-GPT gets the SoTA performance under different evaluation metrics. It achieves 44.17% on Uncod scores while the base image generation model, Stable Diffusion, achieves only 18.81%. Control-GPT also outperforms DALL-E 2, a proprietary OpenAI model with an Uncod score of 37.89%. Control-GPT also achieves SoTA on other scores on Cond compared to other diffusion model variants. Our fine-tuned model can largely translate the performance gain from the programmatic sketches and mitigate the gap between the training and testing data.

## 5.2 FOLLOWING OBJECT POSITION, SIZE AND COLOR IN TEXT PROMPTS

We extend the evaluation to further study more fine-grained control on object sizes and positions specified in text prompts. For evaluation, we randomly sample 100 samples from the Visor dataset and associate them with the position randomly chosen from 4 options and size from 3 options. Figure 5 shows our prompt for querying GPT-4. Detailed setup is presented in the Appendix.

> Draw a giraffe to the left of an apple. The entire image should be 5.12*5.12. The giraffe should be centered at the position (1.5, 2.5) of size (1.0, 1.0). The apple should be centered at position (3.5, 2.5) of size (0.5, 0.5).

Figure 5: **Example prompt for controlling object positions and sizes**. Example prompt for benchmarking object position and size for different models. This is directly passed to the GPT-4 to draw the sketch or ControlNet/Stable Diffusion for generating an image.

**Quantitative evaluation.** Following the convention in Gokhale et al. (2022), we use OWL-ViT to measure whether the model follows the text instruction on the objects' size and location. We run the object detection model for each image and measure whether the detected object size and position match the ones specified in the prompt. Note that we tolerate $\epsilon\%$ error relative to the image size (512×512) for both size and position. If the detected size and position are within $\epsilon\%$ absolute distance relative to the entire figure size, we consider it a success. For baselines, we compare with ControlNet and Stable Diffusion. The ControlNet takes in the same GPT-4 TikZ files and prompts with location and size. Stable diffusion only takes in detailed prompts.

We present the quantitative evaluation results in Table 3. We can see that our Control-GPT model can better control the size and location of the objects given some specifications. Compared to Stable Diffusion (`SD-v1.5`), which hardly has control over object positions and sizes, we improve the overall accuracy from 0% to 14.18%. Compared to the off-the-shelf ControlNet, Control-GPT also achieves better performance across all metrics, obtaining an overall improvement from 8.46% to 4.18%. These results show the potential of our LLM-integrated framework for more fine-grained and precise control of the image generation process.

| | Accuracy | Obj1 Pos | Obj1 Size | Obj2 Pos | Obj2 Size | All Pos | All Size | Pos & Size |
|---|---|---|---|---|---|---|---|---|
| | SD-v1.5 | 1.84 | 1.84 | 2.57 | 2.20 | 1.84 | 2.20 | 0.00 |
| $\epsilon = 3.9\%$ | ControlNet | 13.60 | 15.07 | 15.44 | 15.07 | 15.07 | 15.07 | 8.46 |
| | Control-GPT (ours) | **16.42** | **21.64** | **17.91** | **23.13** | **21.64** | **23.13** | **14.18** |
| | $\Delta$ (vs. SD) | 14.58 | 19.80 | 15.34 | 20.93 | 19.80 | 20.93 | 14.18 |

Table 3: **Results on controlling object positions and sizes.** Control-GPT outperforms the base Stable Diffusion model and the off-the-shelf ControlNet model by a large margin under all metrics. Still, we can see the overall accuracy of fine-grained control over object positions and sizes is relatively low, and our work shows the potential for using LLMs to improve precise control of image generation.

**Visualization.** We additionally present the qualitative results in Figure 6, where we can see Control-GPT can draw objects following the specification of object positions and sizes. In contrast, ControlNet

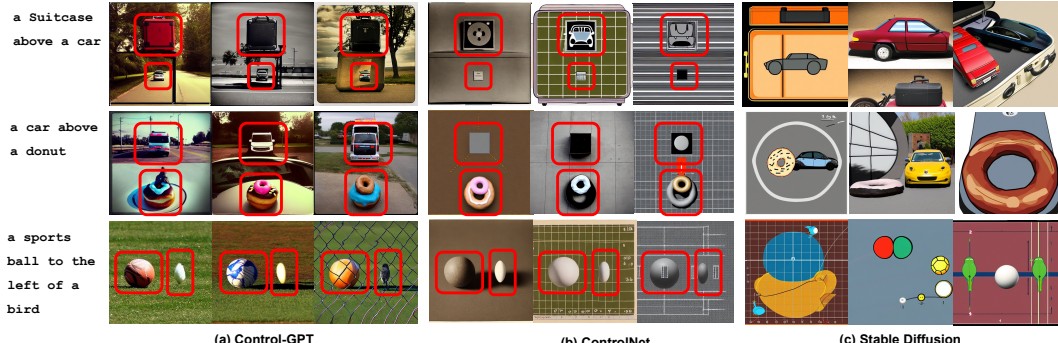

Figure 6: **Controlling object positions and sizes**. Images generated by Control-GPT can follow exactly along the specification of position and size. ControlNet can also follow but struggles to generate the correct object, and Stable Diffusion can't follow the specifications.

| Model | Visor Score (%) | | | | Object Acc (%) | | | |
|---|---|---|---|---|---|---|---|---|
| | left | right | above | below | left | right | above | below |
| GLIDE | 57.78 | 61.71 | 60.32 | 56.24 | 3.10 | 3.46 | 3.49 | 3.39 |
| GLIDE + CDM | 65.37 | 65.46 | 59.40 | 59.84 | 12.78 | 12.46 | 7.75 | 7.68 |
| CV2 | 68.50 | 68.03 | 63.72 | 62.51 | 20.34 | 19.30 | 17.71 | 16.54 |
| DALL-E 2 | 56.47 | 56.51 | 60.99 | 63.24 | **64.30** | **64.32** | **65.66** | **61.45** |
| SD | 64.44 | 62.73 | 61.96 | 62.94 | 29.00 | 29.89 | 32.77 | 27.8 |
| SD + CDM | 69.05 | 66.52 | 62.51 | 59.94 | 23.66 | 21.17 | 23.66 | 24.61 |
| GLIGEN | 36.90 | 36.37 | 36.64 | 33.05 | 19.78 | 19.63 | 21.20 | 18.77 |
| Control-GPT | **72.50** | **70.28** | **67.85** | **65.70** | 49.80 | 48.27 | 47.97 | 46.95 |

Table 4: **More analysis on different spatial relations**. We compare our model with the aforementioned baselines on four variants of spatial relations. Control-GPT consistently outperforms all the baselines in Visor scores. In terms of object accuracy, it also reaches dominant performance except for DALL-E 2.

can also follow but struggles to generate the correct objects and Stable Diffusion cannot follow the specifications.

### 5.3 ABLATION STUDY AND ANALYSIS

**Ablation on spatial relations.** We also study whether the model has a preference for different types of spatial relationships (e.g., left/right/above/below) as part of the analysis for the spatial relation benchmark in Section 5.1. As we can see from Table 4, Control-GPT works consistently better than all the baseline models in terms of Visor Score and object accuracy. One exception is the DALL-E 2 model, which is possibly trained on private data with higher-quality object categories and, therefore, good at generating different objects compared to the rest of open-sourced models. We also notice that text-to-image models often struggle more with `above` and `below` compared to `left` and `right`.

**Relationship between multiple objects.** In previous sections, we have shown evaluations of generating spatial relations of two objects. In this section, we conduct a further evaluation on benchmarking Control-GPT capability of generating multiple objects, with their spatial relationship specified by prompts. We show some qualitative examples in Figure 7. We see that Control-GPT exhibits better performance in understanding the spatial relationship between different objects and putting them in the layout with the help of GPT-4. While DALL-E 2 and Stable Diffusion are often missing generating some objects or wrongly layout them in the figure. One example can be seen as `a sandwich below a TV with a spoon to the right of the sandwich`, where DALL-E 2 and Stable Diffusion fail to generate all the objects, but Control-GPT can manage to do it. The experiment shows the potential of Control-GPT in handling complex scene generations.

**Controllability vs. photo-realism.** We notice that there is often a tradeoff between generating photo-realistic images versus following the exact layout, especially for out-of-distribution text prompts. As shown in Figure 8, the subfigure (a) is an example that the generated image follows exactly the layout, but this results in some artifacts in the image. While in (b), the photo tends to look realistic

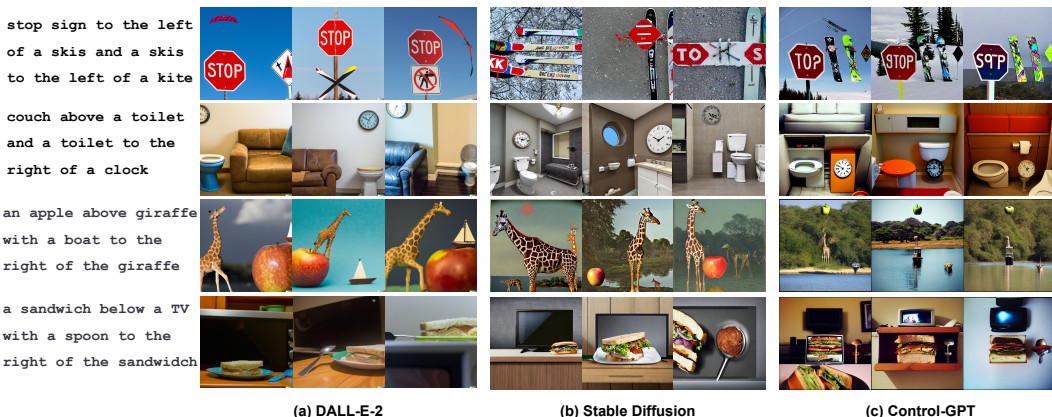

Figure 7: **Image generation with multiple objects**. Control-GPT is better than DALL-E 2 and Stable Diffusion in generating complex scenes involving multiple objects. The two baseline model either suffers from object spatial location misinterpretation or fail to generate the corresponding object.

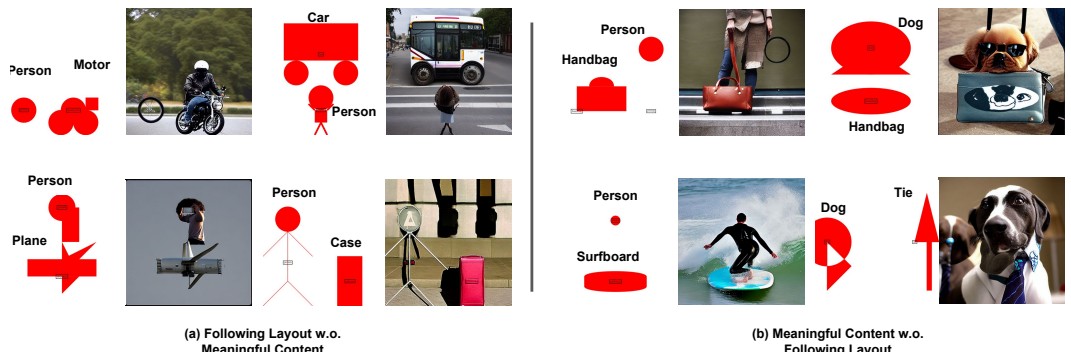

Figure 8: **Controllability vs. photo-realism** . One interesting phenomenon is that Control-GPT has a balance of generating images that follow the guideline versus producing images that look photorealistic. (a) Some examples are that the generated image follows exactly the layout but suffers from artifacts. (b) Examples that look photorealistic but doesn't follow the guideline well.

but doesn't follow the sketch well. We notice this phenomenon in Control-GPT, which leads to the majority of the spatial errors in the experiments. Since Control-GPT tries to balance the photo-realism and instruction following, it explains why Control-GPT didn't fully translate the performance gain of the programmatic sketches ($\sim 97\%$).

## 6 CONCLUSION

We present a novel method Control-GPT, a controllable text-to-image generation method with the guidance of programmatic sketches generated by GPT-4. We augment the ControlNet architecture by grounding tokens and training it with polygons from the existing dataset. By leveraging GPt-4 for generating TikZ sketch and grounding tokens, our method achieves state-of-the-art performance on benchmarks focusing on the spatial locations of different objects. It also demonstrates great potential in controlling object size/position and generating complex scenes. This has large implications for using text-to-image models in many more situations, such as ones where a greater need for creative and editorial control is needed (for example in arts and other creative applications). The paper is also the first to demonstrate a possible way for joint optimization over different AI models, opening up opportunities in the domain.

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
