# CONTROLLABLE TEXT-TO-IMAGE GENERATION WITH AUTOMATIC SKETCHES *Supplementary Materials*

# 1 APPENDIX

## 1.1 NETWORK ARCHITECTURE

In this section, we delve into the intricate details of our network architecture, modeled after the ControlNet architecture. In adhering to the design principles of ControlNet, we implement a unique strategy where all the weights associated with Stable Diffusion are "frozen" or fixed, meaning they do not undergo any changes during the training phase.

But our design doesn't stop there. We've added an innovative component - a controlling branch - that runs parallel to the main network. This supplementary branch is not merely a structural addition; it adds complexity and control to the overall architecture. It allows for intricate manipulations and adjustments that can substantially influence the overall performance and functionality of the network.

By integrating these elements, we believe our architecture balances stability (through freezing Stable Diffusion weights) and adaptability (through the controlling branch), creating a versatile and robust network model.

Table 1: Control-GPT Structure

| Network Block Name | Resolution | Number of Blocks |
|---|---|---|
| SD Encoder Block_1 | 64x64 | 3 |
| SD Encoder Block_2 | 32x32 | 3 |
| SD Encoder Block_3 | 16x16 | 3 |
| SD Encoder Block_4 | 8x8 | 3 |
| SD Middle Block | 8x8 | 1 |

### 1.1.1 BOUNDING BOX EMBEDDING

In this section, we elucidate our bounding box embedding process, which aligns with the methodology detailed in the GLIGEN work. The fundamental strategy we adopt involves the use of Fourier embedding to transform the bounding boxes, thereby enabling more nuanced interaction with the network architecture.

To give a holistic view of the process, consider the bounding box. Rather than including extensive details, we have chosen to only incorporate the central position of the object within each bounding box. This results in a set of pairs representing the x and y coordinates for each object's center denoted as $[X_{obj1}, Y_{obj1}, X_{obj2}, Y_{obj2}, ..., X_{objk}, Y_{objk}]$.

Each pair from this set is then individually processed through the Fourier embedding network. This network serves a role analogous to a tokenizer in a language model, transforming each input into a form that can be effectively processed and interpreted by the subsequent layers of our architecture.

Following this Fourier embedding process, we append the resulting vector representations to the object name embeddings. The concatenated vector, representing spatial information and the object identity, is fed into the higher-level attention network. This multi-layered approach ensures that our model fully comprehends and effectively leverages the rich information within the bounding boxes and associated object identities.

### 1.1.2 OBJECT NAME EMBEDDING

In this segment, we illuminate how the object names associated with each bounding box are embedded. To accomplish this, we rely on the tokenizer and language model encoder from the original stable diffusion model. Given its inherent design to interpret and encode text data, this model offers an organic and efficient means of creating text embeddings for the object names.

Following the text embedding, our process doesn't halt. We move forward to concatenate these text embeddings with two other crucial components - the bounding box embeddings and the image patch embeddings. Each of these elements adds an additional layer of information, integrating spatial data from the bounding box and visual data from the image patch with the textual data from the object name.

Once the concatenation process is complete, we have a multi-dimensional vector that encapsulates a comprehensive range of data. This enriched vector representation is then supplied to the attention network. This network is designed to parse this multi-faceted input, emphasizing important features while downplaying less critical information. Through this comprehensive and layered approach, our model ensures a thorough understanding and utilization of the diverse data it is provided with.

### 1.1.3 TOKEN FUSION

We simply concat all the tokens before feeding them to the ControlNet structure.

$$\text{Embedding} = \text{Concat}([l_{\text{bbox}}, l_{\text{name}}, l_{\text{image}}]) \tag{1}$$

## 1.2 DATASET DETAILS

In refining our model, we employ the LVIS dataset, a richly annotated subset of the COCO dataset, renowned for its broad array of image categories and detailed labels. This dataset forms the bedrock of our fine-tuning process, as we incorporate all the images from the training and evaluation segments.

The depth of our dataset extends beyond mere images. We also utilize the bounding boxes, object names, and polygons provided within the LVIS dataset. Each component adds a layer of richness and complexity to our data. The bounding boxes provide spatial information, indicating the position and extent of each object within the images. The object names offer categorical data, identifying the type of object encapsulated by each bounding box. The polygons provide even more granular spatial data, detailing the precise shape and orientation of the objects.

By combining these diverse data types – image, spatial, categorical, and shape – we construct a dataset that is not only varied but also highly detailed. This enriched dataset allows us to fine-tune our model more effectively, optimizing its performance across a wide array of scenarios.

## 1.3 TRAINING DETAILS

For training, we use the default training parameters as ControlNet. Detailed hyperparameters are listed below in Table 2.

| Hyperparameter Name | Value |
| --- | --- |
| batch size | 8 |
| learning rate | 1e-5 |
| sd locked | True |
| only mid control | False |
| Fourier Embedding Dim | 16 |
| Object Embedding Dim | 768 |

Table 2: Control-GPT hyperparameters

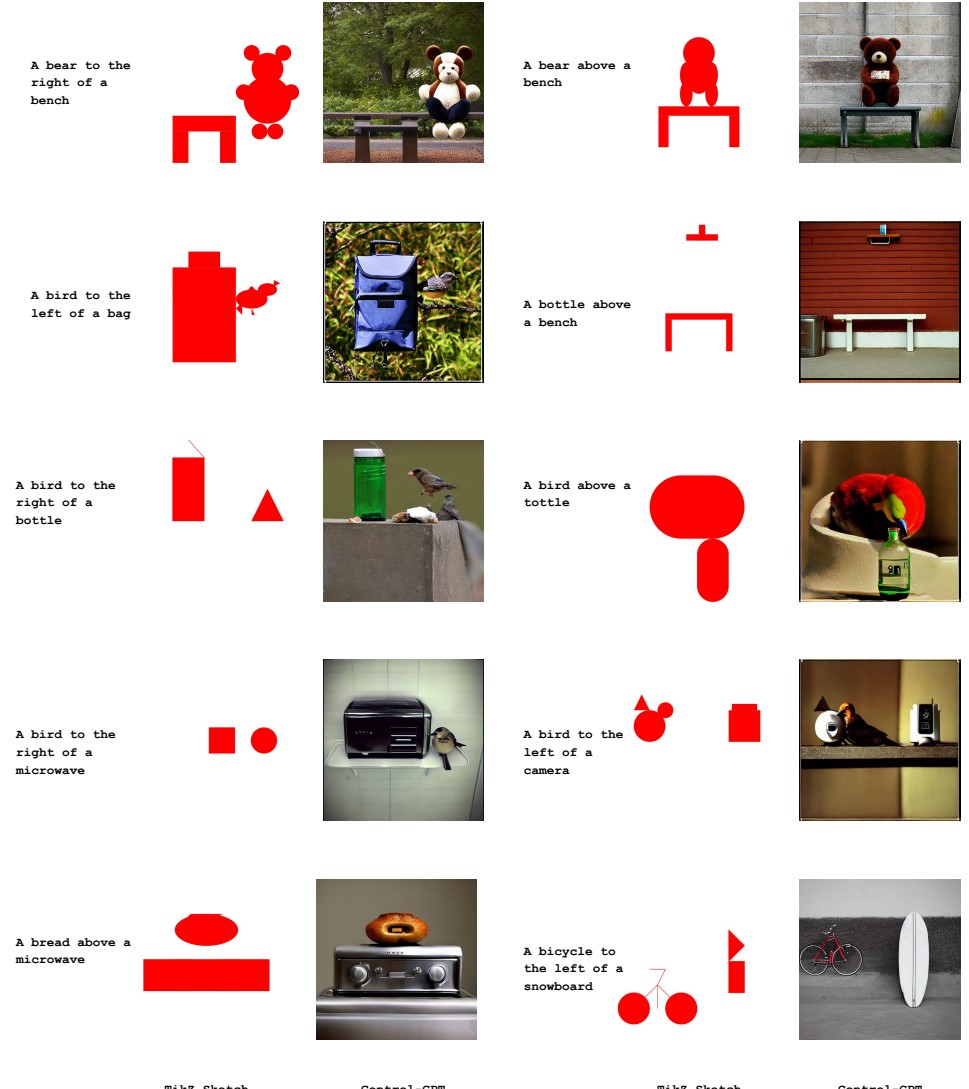

Figure 1: More sketch examples Control-GPT in the spatial relation dataset. Control-GPT is being able to generate images very precisely without any missing objects. It also balances the visual quality and layout following to make the image look vivid.

## 1.4 MORE EXAMPLES FOR CONTROL-GPT

To further illustrate our points, we present additional examples derived from Control-GPT, a modification of GPT-4 designed to enhance precision in control. These examples are visually depicted in Figure 1 and Figure 2 for your perusal.

Interestingly, the TikZ sketch associated with the GPT-4 model doesn't necessarily offer comprehensive control over the output. This presents a unique opportunity for Control-GPT. In response to this gap, Control-GPT has been devised to strike a fine balance between two pivotal factors: the quality of the generated image and the adherence to the prescribed layout.

By navigating this delicate balance, Control-GPT excels at generating high-quality images that closely follow the given layout specifications. In essence, Control-GPT augments the core functionality of GPT-4, enhancing control without compromising image quality, thereby offering an upgraded solution for more precise and aesthetically pleasing outputs.

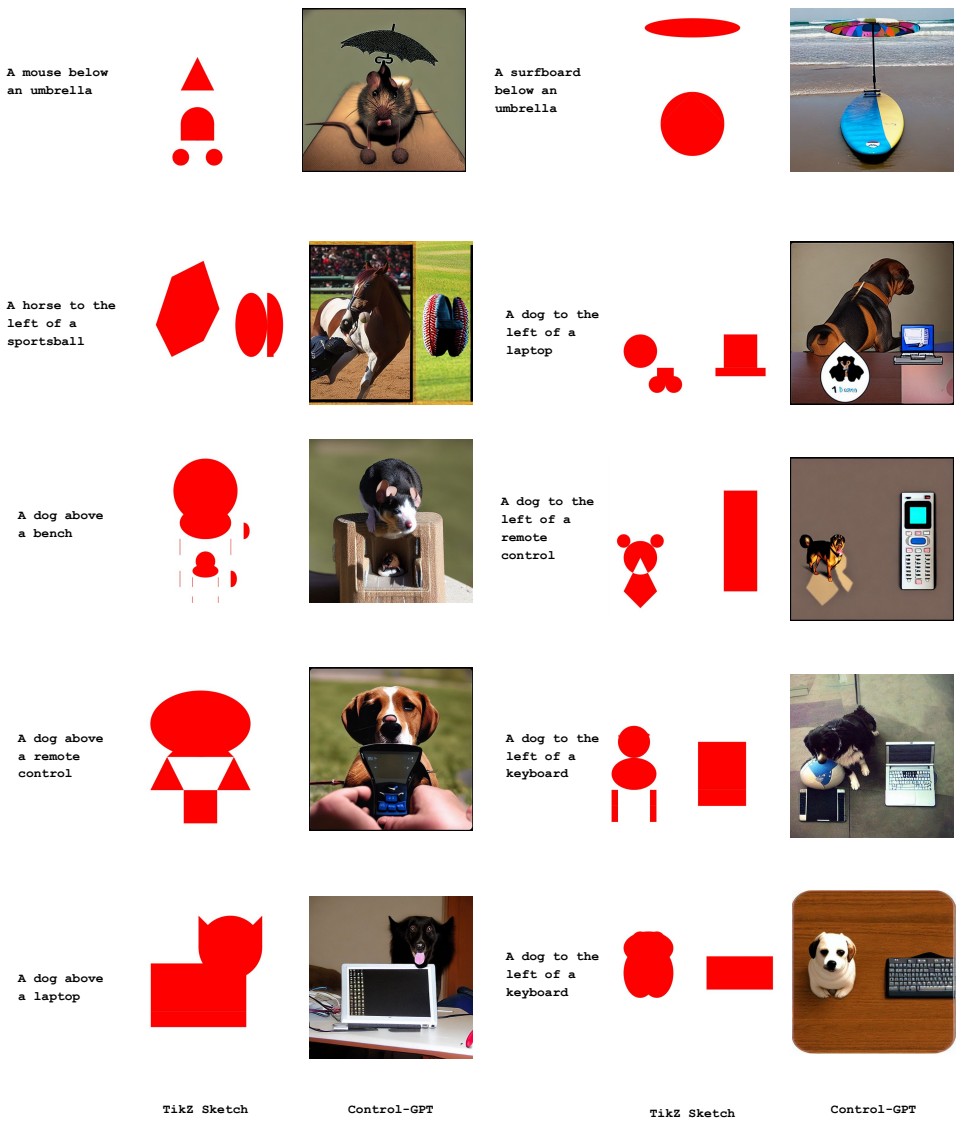

Figure 2: Control-GPT is able to generate images very precisely without any missing objects in the spatial relation dataset. It also balances the visual quality and layout following to make the image look vivid.

## 1.5 TikZ Code Snippets

We also show some code snippets in Figure 3, which are entirely generated by GPT-4. We see that the code structure is almost correct with proper comments. The code compiles without any error. This is quite impressive. In addition to that, it captures all the necessary components. This demonstrates GPT-4's strong capability of drawing TikZ code.

## 1.6 Vanilla ControlNet Examples

In addition, we have conducted an experiment on directly passing the TikZ images to the vanilla ControlNet trained on segmentation maps. We found that the Vanilla ControlNet is being able to follow the outline of the sketch very well. This is not surprising as it is trained to do this. However, it doesn't understand the concept of the image well as it almost fails to generate all the objects specified in the prompt. The results in Figure 4 demonstrate some examples. ControlNet rarely generates the correct objects.

## 1.7 More TikZ Examples

We also plot more examples from different LLMs we tested in Figure 5. GPT-4 demonstrates strong capability in plotting more details in the figure, including all the objects, and getting their spatial location correct. Compared to that, GPT-3.5 and Chat-GPT tend to either miss some objects or don't use enough details in the figure.

## References

```
\begin{tikzpicture}
\path[use as bounding box] (0,0) rectangle (5.12,5.12);
% Draw the person
\fill[red] (1,1) circle (0.5);
\fill[red] (1,0.5) rectangle (1.1,1.5);
\fill[red] (1.1,1.5) – (1.5,2.5) – (1.7,1.5) – cycle;
\fill[red] (1.1,0.5) – (1.5,-0.5) – (1.7,0.5) – cycle;
% Draw the bus
\fill[red] (3,1) rectangle (4.5,2.5);
\fill[red] (3.25,1.25) circle (0.25);
\fill[red] (4.25,1.25) circle (0.25);
\end{tikzpicture}
```

```
\begin{tikzpicture}
\path[use as bounding box] (0,0) rectangle (5.12,5.12);
% Truck
\fill[red] (0.5,1) rectangle (3,2.5);
% Truck body
\fill[red] (2.5,2.5) rectangle (3.5,3.5);
% Truck cabin
\fill[red] (1,1) circle (0.5);
% Wheel 1
\fill[red] (2.5,1) circle (0.5);
% Wheel 2
\fill[red] (3.5,1) circle (0.5);
% Person
\fill[red] (4,3) circle (0.5);
% Head
\fill[red] (3.75,1.5) rectangle (4.25,2.5);
% Body
\draw[red, line width=2pt] (4,2.5) – (3.5,1);
% Leg 1
\draw[red, line width=2pt] (4,2.5) – (4.5,1);
% Leg 2
\draw[red, line width=2pt] (4,3) – (3.5,3.5);
% Arm 1
\draw[red, line width=2pt] (4,3) – (4.5,3.5);
% Arm 2
\end{tikzpicture}
```

```
\begin{tikzpicture}
\draw[red, fill=red] (1,2) circle (0.25);
% head
\draw[red] (1,2) – (1,1);
% body
\draw[red] (1,1.5) – (0.5,1.5);
% left arm
\draw[red] (1,1.5) – (1.5,1.5);
% right arm
\draw[red] (1,1) – (0.5,0.5);
% left leg
\draw[red] (1,1) – (1.5,0.5);
% right leg
\draw[red, fill=red] (3.5,0.5) – (4.5,0.5) – (4.12,1) – (3.88,1) – cycle;
% boat
\useasboundingbox (0,0) rectangle (5.12,5.12);
\end{tikzpicture}
```

Figure 3: Examples of TikZ code GPT-4 generates. It almost gets all the syntax correct: the code compiles without any error in LateX.

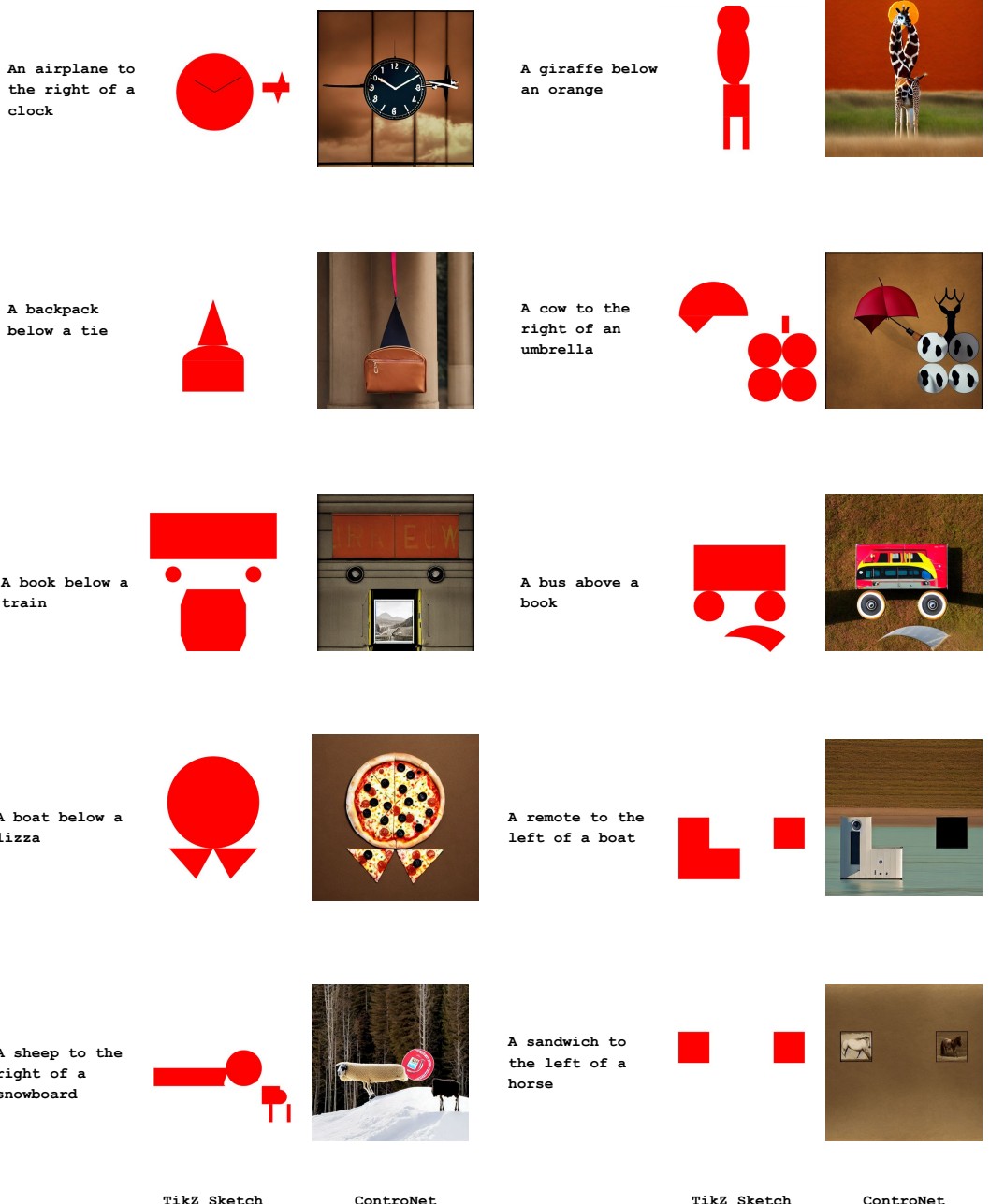

Figure 4: More examples from the vanilla ControlNet trained on segmentation maps. It hardly generates the correct objects, as we see in the figure. Despite that, its ability to follow the outline sketch is pretty good.

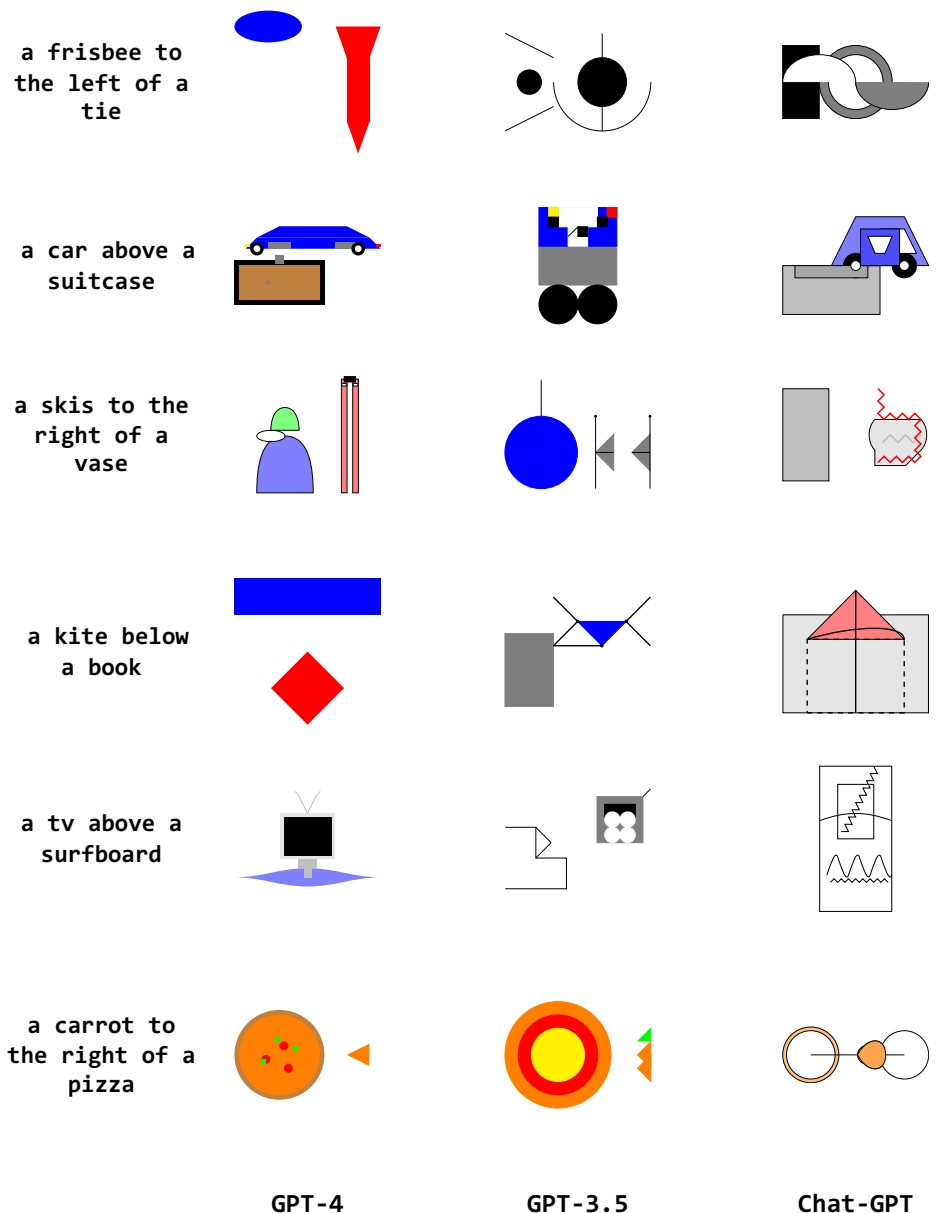

Figure 5: More sketch examples from GPT-4, GPT-3.5, and ChatGPT. GPT-4 consistently outperforms other models in following text instructions and understanding the spatial relations