# OpenReview forum: "Controllable Text-to-Image Generation with Automatic Sketches"
_ICLR.cc/2024/Conference — Submitted to ICLR 2024_

### Official Review · Reviewer_nAwp · 2023-11-01

**Soundness:** 2 fair
**Presentation:** 2 fair
**Contribution:** 1 poor
**Rating:** 5
**Confidence:** 5

**Summary:**

The authors introduce Control-GPT, which is claimed as a simple yet effective framework that leverages LLMs to generate sketches based on text prompts firstand thereafter feed those sketches to Control-GPT to generate images. Satisfactory qualitative results justify the efficiency of the method.

**Strengths:**

1. The concept introduced here is really interesting, although the components used here carry less novelty.

2. The writing of introduction, and the overall paper is quite fluid and easy to understand.

3. Providing additional baselines apart from state-of-the-arts competitors is appreciated.

**Weaknesses:**

1. Although the authors frequently use the term 'sketches', they are not really sketches ([D], [E]), rather quite far from it in that they do not model human subjectivity at all.

2. This paper seems more of a combination of multiple concepts like Diffusion models + GPT + conditioning signal encoder. Although there can be novelty depending upon the formulation of usage of pre-known concepts, this paper lacks a strong formulation in that aspect.

3. Given there are papers which can directly generate the same quality images from sparse line drawings (e.g., ControlNet [A], T2I-Adapter [B], Voyonov et al. [C], etc.), it is not very appealing to have yet one more paper that generates additional pseudo sketches to generate image. As in -- why should I write a 100 word prompt to generate a "pseudo sketch" and then generate an image from it?

4. Certain claims like "The paper is also the fist to demonstrate a possible way for joint optimization over different AI models" are quite strong without adequate proof or citations.

5. Quality of Figures could be improved.

6. Although technical aspects are there in this paper, the motivation or the need is not quite clear.

Minor:
Typo: Conclusion section GPt-4 > GPT-4

[A] Adding Conditional Control to Text-to-Image Diffusion Models. Lvmin Zhang, Anyi Rao, Maneesh Agrawala.
[B] T2I-Adapter: Learning Adapters to Dig out More Controllable Ability for Text-to-Image Diffusion Models. Chong Mou, Xintao Wang, Liangbin Xie, Yanze Wu, Jian Zhang, Zhongang Qi, Ying Shan, Xiaohu Qie
[C] Sketch-Guided Text-to-Image Diffusion Models. Andrey Voynov, Kfir Aberman, Daniel Cohen-Or.
[D] How Do Humans Sketch Objects?. Mathias Eitz, James Hays and Marc Alexa.
[E] Why Do Line Drawings Work? A Realism Hypothesis. Aaron Hertzmann

**Questions:**

1. What is necessity of TiKz figures here?
2. Why do I need to write a lengthy prompt? What is that significant a necessity for prompt which can not be acquired from directly drawing or sketching a semantic map?

---

### Official Review · Reviewer_4EN4 · 2023-11-04

**Soundness:** 3 good
**Presentation:** 3 good
**Contribution:** 2 fair
**Rating:** 3
**Confidence:** 4

**Summary:**

To enhance controllable text-to-image generation, the authors propose using ChatGPT to initially generate TikZ sketches. These sketches are then fed into a fine-tuned ControlNet. Furthermore, to facilitate the fine-tuning of ControlNet, the authors adapt existing datasets to create aligned text, images, and polygons that mimic the sketches. Experiments demonstrate the effectiveness of the proposed method in terms of controllability.

**Strengths:**

1. The writing is clear and easy to follow.

2. The authors combine Large Language Models and Diffusion Models to achieve controllability in text-to-image generation.

3. The experiments provide evidence that the proposed method can effectively control the generation of objects that match the given text, in terms of position and also size.

**Weaknesses:**

1. The proposed method relies on existing models, including ChatGPT and ControlNet, which limits its technical novelty.

2. The authors suggest using ChatGPT to produce TikZ sketches and then fine-tuning ControlNet with TikZ sketches, text, and grounding tokens. However, pretrained ChatGPT can accept semantic segmentation as input, and ChatGPT can produce code to generate the required mask with the desired position and size. In light of this, why don't the authors simply use the semantic mask, which would eliminate the need for further fine-tuning of ControlNet?

3. I am confused about the quantitative comparison presented in Table 3 and the qualitative comparison in Figure 6. What is the experimental setting for ControlNet? What inputs does ControlNet receive? Additionally, in Table 2, the authors only compare their method with image generation models without additional semantic inputs. Why don't the authors include ControlNet in this comparison? I am also puzzled by the baseline; the authors claim it includes ControlNet with Segmentation and ControlNet with Canny Detection, but I cannot find these two baselines in the experiments.

4. In Figure 3, the top example appears to depict two giraffes rather than two zebras.

**Questions:**

See above weaknesses.

---

### Official Review · Reviewer_uyRt · 2023-11-06

**Soundness:** 2 fair
**Presentation:** 3 good
**Contribution:** 2 fair
**Rating:** 5
**Confidence:** 4

**Summary:**

This work proposes to use an LLM to produce TikZ code to generate sketches from text. The sketches and text can then guide a ControlNet-like architecture for layout-to-image synthesis focusing on correct spatial arrangement.

**Strengths:**

- The paper tackles an important problem, namely to improve spatial instruction following in current text-to-image models, which tend to ignore fine-grained information such as spatial arrangement, object relationships, numerical information and attribute binding.
- The paper is well-written, with a clear presentation of ideas, experiments, and results.
- To my knowledge, conditioning image generative models on TikZ sketches is novel.
- The results suggest that the model can better follow the spatial instructions provided via text.

**Weaknesses:**

- The paper should identify the advantages of using TikZ over simpler geometric shapes like rectangles or leveraging text-to-layout/mask prediction networks. There has been prior work using LLMs to generate layouts (e.g., https://arxiv.org/abs/2305.15393, https://arxiv.org/abs/2308.05095).

- The dataset creation process, which involves extracting polygons from COCO, raises the question of why not use the instance masks (or bounding boxes) directly as a baseline, which could eliminate the need for an LLM, TikZ code generation, and image compilation. In fact, fine-tuning on instance masks from COCO should be considered a baseline to analyze the possible benefits of using TikZ sketches. Sampling can be realized through a additional text-to-layout/mask prediction network (many prior GAN methods explored this).

- There seems to be a lack of fine-tuning the vanilla ControlNet on the TikZ dataset, which should be considered as an important baseline for the study.

- The vanilla ControlNet baseline is missing in table 2, and table 4.

- Several of the generated TikZ sketches presented in the appendix are weird:
    - appendix fig1:
        - A bird to the left of a bag is spatially wrong
        - A bird above a (b?)tottle has a weird bird shape and could be replaced by a rectangle
    - appendix fig2:
        - A dog to the left of * ; all are different and the generated image does not follow the masks

- Missing references for the models listed in table 2 of the paper.

- ControlGPT is incorrectly identified as the best in the Visor 2 metric when it is actually Dalle-2 that has the highest score.

- Figures 7 in the main paper and figures in the appendix should visualize generated images from ControlNet baseline as well as ControlGPT side-by-side for a more comprehensive evaluation of visual results.

**Questions:**

- Why prompting GPT to fill each object with red color and keeping objects separated? Wouldn't using different colors for different objects be more sensible, and how are overlapping or occlusion scenarios between objects managed.

- Dalle-2 shows better performance in object accuracy according to table 2, yet it scores lower in the Visor metric. The paper should include a discussion on the possible reasons behind this discrepancy.

---

### Meta-Review · Area_Chair_jc2i · 2023-12-10

**Metareview:**

This paper was reviewed by three knowledgeable referees. The main concerns raised by the reviewers included:
1. The motivation of the proposed approach (and use of TikZ sketches) appeared questionable (uyRt, 4EN4, nAwp)
2. The experimental evidence appeared unconvincing (e.g. missing important baselines) (uyRt, 4EN4)
3. The novelty appeared incremental (combination of existing methods) (4EN4, nAwp)
4. The claims made appeared too strong given the experimental evidence (nAwp).

Unfortunately there was no rebuttal and the concerns raised by the reviewers remain. Therefore, the AC recommends to reject and encourages the authors to consider the feedback provided by the reviewers to improve future iterations of their work.

**Justification For Why Not Higher Score:**

The reviewers raised important concerns and there was no rebuttal to address them.

**Justification For Why Not Lower Score:**

N/A

---

### Decision · Program_Chairs · 2024-01-16

Reject